# Characterization of Three Somatic Mutations in the 3′UTR of *RRAS2* and Their Inverse Correlation with Lymphocytosis in Chronic Lymphocytic Leukemia

**DOI:** 10.3390/cells12232687

**Published:** 2023-11-22

**Authors:** Marta Lacuna, Alejandro M. Hortal, Claudia Cifuentes, Tania Gonzalo, Miguel Alcoceba, Miguel Bastos, Xosé R. Bustelo, Marcos González, Balbino Alarcón

**Affiliations:** 1Immune System Development and Function Program, Centro Biología Molecular Severo Ochoa, Consejo Superior de Investigaciones Científicas (CSIC)-Universidad Autónoma de Madrid, 28049 Madrid, Spain; marta.lacuna-mohedano20@imperial.ac.uk (M.L.); ahortal@cbm.csic.es (A.M.H.); c.cifuentes@cbm.csic.es (C.C.); tania.gonzalo@estudiante.uam.es (T.G.); 2Departamento de Hematología, Hospital Universitario de Salamanca (HUS-IBSAL), 37007 Salamanca, Spain; alcocebasanchez@saludcastillayleon.es (M.A.); mbastos.ibsal@saludcastillayleon.es (M.B.); margondi@usal.es (M.G.); 3Centro de Investigación del Cáncer, Instituto de Biología Molecular y Celular del Cáncer and Centro de Investigación Biomédica en Red de Cáncer, CSIC, Universidad de Salamanca, 37007 Salamanca, Spain; xbustelo@usal.es

**Keywords:** *RRAS2*, somatic mutations, 3′ untranslated region, Chronic Lymphocytic Leukemia (CLL), clinical implications

## Abstract

Chronic lymphocytic leukemia (CLL) is a hematologic malignancy characterized by progressive accumulation of a rare population of CD5+ B-lymphocytes in peripheral blood, bone marrow, and lymphoid tissues. CLL exhibits remarkable clinical heterogeneity, with some patients presenting with indolent disease and others progressing rapidly to aggressive CLL. The significant heterogeneity of CLL underscores the importance of identifying novel prognostic markers. Recently, the RAS-related gene *RRAS2* has emerged as both a driver oncogene and a potential marker for CLL progression, with higher *RRAS2* expression associated with poorer disease prognosis. Although missense somatic mutations in the coding sequence of *RRAS2* have not been described in CLL, this study reports the frequent detection of three somatic mutations in the 3′ untranslated region (3′UTR) affecting positions +26, +53, and +180 downstream of the stop codon in the mRNA. An inverse relationship was observed between these three somatic mutations and *RRAS2* mRNA expression, which correlated with lower blood lymphocytosis. These findings highlight the importance of *RRAS2* overexpression in CLL development and prognosis and point to somatic mutations in its 3′UTR as novel mechanistic clues. Our results may contribute to the development of targeted therapeutic strategies and improved risk stratification for CLL patients.

## 1. Introduction

B-cell chronic lymphocytic leukemia (CLL) is a mature B-cell neoplasm characterized by the progressive accumulation of a rare population of mature B-lymphocytes (classified as CD19+CD5+) in the blood and lymphoid organs [1]. It is the most prevalent type of leukemia in the Western world with an overall incidence rate of 4.2 per 100,000 [2,3]. CLL presents a gender disparity of approximately 2:1, with a higher risk associated with the male faction (6.1 per 100,000 in males compared to 3.1 in females) [4,5]. In the United States alone, approximately 18,740 new cases of CLL are diagnosed each year (American Cancer Society). This comprises roughly 25–30% of all newly diagnosed leukemia cases and accounts for about 1.1% of all new cancer diagnoses. CLL primarily affects older individuals, with the typical age of diagnosis hovering at approximately 70 years [2]. The incidence of CLL rises notably after the age of 50 and continues to increase with advancing age [6].

While increasing age is one of the most significant risk factors for CLL, certain genetic abnormalities have been associated with an increased risk of developing the disease. Cytogenetic studies reveal recurrent chromosomal alterations, including deletions at loci 11q, 13q, 17p, and trisomy 12 [7]. Moreover, CLL patients with an unmutated IGHV status have a less favorable prognosis. Unmutated IGHV is associated with a more rapid disease progression, a shorter time for treatment, and reduced overall survival rate. In these patients, the IGHV gene has not undergone somatic hypermutation, suggesting that the precursor of the leukemic cells is a pre-germinal center (GC) B-cell less terminally differentiated than that of CLL with mutated IGHV [7]. Although CLL has been extensively studied, the molecular mechanisms underlying disease development are still to be fully elucidated. Genes deregulated and/or altered in CLL have been identified and singled out as possible driver genes. This includes genes such as *TP53*, *ATM*, *MYD88*, *NRAS*, and *KRAS*, which are involved in fundamental cellular processes such as DNA repair, cell-cycle control, and Notch, Wnt, and B-cell receptor (BCR) signaling [8].

RAS proteins are a family of small guanosine triphosphate hydrolases (GTPases) that include well-known oncogenic players such as aforementioned K-RAS and N-RAS, alongside H-RAS. R-RAS2 belongs to the RAS-related subfamily of RAS proteins, which share approximately 60% amino acid identity with classic RAS. Seemingly, R-RAS2 also shares associated proteins that regulate its activation–inactivation cycles with its classical counterparts including guanine nucleotide exchange factors (GEFs) and GTPase-activating proteins (GAPs) [9,10]. R-RAS2 has been identified as a key player in immune system homeostasis. It interacts with both the B- and T-cell receptors (BCR and TCR, respectively), facilitating the generation of tonic survival signals from these receptors [11]. In B-cells, R-RAS2 has been shown to be crucial for the effective establishment of a proper GC reaction by regulating B-cell metabolism [12].

Although early studies from the 1990s show that oncogenic mutations in *RRAS2* (G23V, Q72L) have higher transformation potential than analogous mutations in classical *RAS* genes (G12V, Q61L), *RRAS2* is mainly found overexpressed in its wild-type form rather than mutated [9,10]. It has been described that R-RAS2 protein is overexpressed in different types of human malignancies, including skin cancers [13], oral cancers [14], and esophageal tumors [15]. We have recently demonstrated a driver role for *RRAS2* bearing the activating mutation Q72L, which causes T-cell acute lymphocytic leukemia, a leukemia of immature B-cells, a Harderian gland adenoma, and an ovarian cystadenocarcinoma in 100% of mice, among other malignancies [16]. However, those mice do not develop leukemias or lymphomas of mature T-cell or B-cell origin. In contrast, in another recent publication, we demonstrated that overexpression of wild-type human *RRAS2* bearing no mutations causes the development of B-cell chronic lymphocytic leukemia in 100% of mice, thus demonstrating that wild-type *RRAS2* is an oncogene driver if overexpressed [17]. The driver role in mice was paralleled by the finding of wild-type *RRAS2* mRNA overexpression in 82% of human CLL blood samples. Moreover, *RRAS2* overexpression was found to be higher in patients with full-blown CLL than in patients with the pre-malignant stage monoclonal B-cell lymphocytosis (MBL). In addition, higher *RRAS2* overexpression was correlated with higher lymphocytosis, a higher percentage of CD19+CD5+ malignant cells in the blood, and a reduced number of platelets, all data indicating that *RRAS2* mRNA expression levels are associated with a more aggressive disease and worse prognosis.

In addition to the association between CLL aggressiveness and *RRAS2* expression in patients, we found another powerful correlation in human disease that sustains our hypothesis suggesting *RRAS2* overexpression is one of the key driver oncogenes in CLL. We found the existence of a single nucleotide polymorphism (SNP) (rs8570) that catalogues the change of the canonical G-nucleotide at position +124 after the stop codon in the *RRAS2* 3′UTR for a C [17]. The presence of a C-allele at the SNP position, whether in homozygosis or heterozygosis, was strongly linked to higher *RRAS2* mRNA expression and various indicators of worse prognosis.

In this paper, we present a comprehensive investigation into the mutations occurring within the 3′UTR of *RRAS2*, revealing a striking pattern of recurrent mutations at positions +26, +53, and +180 after the stop codon. We show how these somatic mutations, distinct from the SNP rs8570, associate with *RRAS2* mRNA expression and markers of CLL aggressiveness.

## 2. Materials and Methods

### 2.1. Human Cells

Human blood samples were collected from volunteer patients with chronic lymphocytic leukemia (CLL) at the Hematology Unit of Salamanca University Hospital. The collection was carried out following the provision of written informed consent, and the study was authorized under the number PI 2019 03217. All participants, including both patients and healthy volunteers, belong to the Caucasian ethnicity. Fresh human peripheral blood mononuclear cells (PBMCs) were obtained through density centrifugation using a Lymphoprep™ (StemCell Technologies, Vancouver, BC, Canada) gradient of whole blood. These PBMCs were utilized for flow cytometry, RT-qPCR analysis, and gDNA extraction.

### 2.2. Flow Cytometry

Human single-cell suspensions were incubated for 15 min with Ghost Dye™ 540 (Tonbo) in PBS to label and discard dead cells from analysis. Cells were washed with PBS + 2% FBS and incubated with fluorescently labelled antibodies for 30 min at 4 °C after. Afterwards, cells were washed in PBS + 2% FBS and data were collected on a FACS Canto II (BD Biosciences, Franklin Lakes, NJ, USA) cytometer. A minimum of 50,000 and a maximum of 200,000 events was acquired in every measurement. Analyses were performed using FlowJo software (vX.0.7, BD Biosciences, Franklin Lakes, NJ, USA).

### 2.3. gDNA Extraction

A total of 10^6^ cells were harvested per sample. Then, 500 μL of lysis buffer (Tris-HCl pH 8 50 mM, NaCl 200 mM, EDTA 10 mM, SDS 1% and fresh proteinase K 0.2 mg/mL) were added to each sample and incubated overnight (ON) at 55 °C. The next day, gDNA was purified using phenol chloroform and resuspended in 100 μL of 10 mM Tris-HCl pH 8.5 depending on the pellet size obtained after the last centrifugation step.

### 2.4. Sequencing Strategy for Patients’ Samples

A total of 10^6^ cells per patient sample were used for RNA extraction from PBMCs of CLL patients. RNA was isolated using the RNAeasy Plus Mini Kit (QIAGEN Sciences, Germantown, MD, USA). cDNA was synthesized with SuperScript III (Thermo Fisher Scientific, Waltham, MA, USA) using Oligo-dT primers. cDNA was used as the template to sequence the 3′UTR region of *RRAS2*. Specific oligonucleotides were used to detect the presence of the canonical or alternative allele in each position (26, 53, 124, 180; Table 1) by quantitative real-time PCR (qPCR). All qPCR readings were performed in triplicate using 100 ng cDNA, SYBR Green PCR Master Mix, and gene-specific primers (Table 1) in an ABI 7300 Real Time PCR System. The oligonucleotides in Table 1 were designed as described in [18]. As a loading reference, specific oligonucleotides aligning to constitutive *RRAS2* exons 3 and 4 were used (forward: GCA GGA CAA GAA GAG TTT GGA; reverse: TCA TTG GGA ACT CAT CAC GA). The obtained cycle threshold (Ct) values were used to normalize *RRAS2* 3′UTR mRNA expression in each individual patient relative to their respective exons 3 and 4 expression. This approach accounted for variability in gene expression across samples, ensuring accurate comparisons.

Table 1 provides a detailed overview of the specific oligonucleotides utilized in the qPCR analysis to characterize mutations within RRAS2 3′UTR. Notably, the nucleotide in blue type represents the mutation site in the somatic mutations 26, 53, and 180 as well as the SNP (rs8570) at 124. The red nucleotides differ from the native gene sequence to enhance the correct separation of the allele populations according to the results in [18].

### 2.5. RRAS2 Expression Measurement

A set of primers that expand constitutive exons 3 and 4 was used to measure mRNA expression of *RRAS2* in patient PBMCs (forward: GCA GGA CAA GAA GAG TTT GGA; reverse: TCA TTG GGA ACT CAT CAC GA). Obtained cycle threshold (Ct) values were used to calculate mRNA levels relative to 18S rRNA expression using the 2^(−ΔΔCt)^ method. Outliers for *RRAS2* expression were identified using ROUT model at Q = 0.1% to remove definite outliers from analysis.

### 2.6. Fluorescent Probes Method to Sequence rs8570 SNP in CLL Patients’ Samples

Fluorescent dual-labelled probe technology, pre-developed by Applied Biosystems (Foster City, CA, USA) and custom made to target rs8570 SNP was used to sequence our CLL and healthy patients. A detailed description of the methodology employed by the custom-made TaqMan technology is available at [19]. All qPCR reactions were performed using an initial cycle of 10 min at 95 °C for polymerase activation, followed by 40 cycles of 15s at 95 °C and 1 min at 60 °C. rs8570 allele distribution was assessed by plotting the relative fluorescent units (RFU) of both fluorophores in a scatter-plot representation.

### 2.7. Statistical Analysis

Statistical parameters including the exact value of n and the mean +/− S.D. or SEM. are described in the Figures and Figure legends. Parametric and non-parametric tests were carried out after assessing the normal distribution of the data. Two-tailed Student’s t-test, ANOVA tests, Mann–Whitney test, Chi-squared test with or without Yates’ correction, and Dunn’s multiple comparisons tests were used as indicated in the Figure legends to assess the significance of mean differences. All data were analyzed using the GraphPad Prism 9.5.1 software (GraphPad Software, Boston, MA, USA).

## 3. Results

### 3.1. Presence of Somatic Mutations in the 3′UTR of RRAS2 in CLL Patients

We have previously shown that the SNP rs8570 in the 3′UTR of human *RRAS2* is associated with higher mRNA abundance and a worse prognosis in CLL patients. Expression of the non-canonical C-allele in one or two dosages in leukemic cells was linked to higher *RRAS2* expression than in leukemic cells carrying two copies of the canonical G-allele (GG) [17]. Furthermore, the frequency distribution of G- and C-alleles at SNP rs8570 was not in Hardy–Weinberg equilibrium [20], suggesting the presence of selective pressure favoring the C-allele within the human CLL patient population. Recently, we developed a new quantitative PCR method to enhance the genotype characterization at the SNP position [19]. Using this new method, we analyzed the frequencies of GG, GC, and CC genotypes at the SNP position in the blood of an extended cohort of CLL patients (n = 203) and compared these frequencies with those of a cohort of healthy blood donors (n = 235) (Figure 1A,B; all sample data is shown in Appendix A). Frequency analysis confirmed that the SNP alleles were not in equilibrium in the CLL sample population (Figure 1A), whereas the two alleles were in equilibrium in the healthy donor population (Figure 1B). Contingency tests of GG, GC, and CC frequencies in the patient and healthy blood cohorts showed that the imbalance in allele distribution was caused by an increased frequency of CC homozygotes in the patient population at the expense of GC heterozygotes (Figure 1C). These results confirmed the idea that the C-allele at SNP rs8570 position in the 3′UTR of *RRAS2* mRNA is enriched in homozygosity in leukemic cells. Since the C-allele is associated with higher mRNA abundance [17], these data further support the idea that overexpression of wild-type *RRAS2* is behind the development of CLL.

In contrast to *KRAS*, the presence of somatic mutations in the coding sequence of RRAS2 is very rare in human cancers. In CLL, a total of 1308 patient samples from four different studies showed no mutations in the coding sequence of *RRAS2* (cBioportal.org). To further characterize the *RRAS2* mRNA, we sequenced the 3′UTR region using the Sanger method. We uncovered additional mutations, alongside that of SNP rs8570, including mutations at positions +26 (G>T), position +53 (T>C), and position +180 (T>C) after the stop codon (Figure 2A). Since none of these alterations have been reported as SNPs in databases, we hypothesize that they arise from somatic mutations in CLL patients. Next, we designed two sets of primer pairs per position to evaluate the frequencies of the canonical and alternate alleles by RT-qPCR (see Section 2). The results of both PCRs were normalized by obtaining the ratio of the alternate allele to the canonical allele. Such a ratio was calculated for each of the collected CLL blood samples and plotted (Figure 2B–D). The median and mean of the alternate/canonical ratios were calculated for each position, and samples with ratios above the median were considered to express the somatic mutation at least in heterozygosity. Interestingly, some blood samples contained somatic mutations in two of the three positions and some in all three positions simultaneously (red dots in Figure 2B–D). Our data showed that 51% of the CLL samples contained at least one mutation at either one of the three positions (Figure 2E). Each of the three positions was mutated with similar frequency (37–38%), whereas 20% of the samples carried two mutations, with the double mutation +26+180 being the most common, and approximately 3% carried mutations at all three positions (Figure 2E). These results show that, in contrast to missense mutations in the *RRAS2* gene, mutations in the 3′UTR at three specific positions are common in CLL.

### 3.2. Association of the Three 3′UTR Mutations with RRAS2 mRNA Abundance

Next, we investigated if there was an association between the presence of mutations at positions +26, +53, and +180 and the allele distribution at the SNP rs8570 position. To this end, we first plotted the alternate/canonical ratios for each position in the CLL cohort classified as GG, GC, and CC SNP genotypes. We found that the ratios in favor of the alternate alleles were higher in samples with GC genotype than in GG and CC homozygotes, although they reached statistical significance only for position +26 when comparing GC and GG genotypes (Figure 3A). Calculation of the average number of mutations at any of the three positions in the 3′UTR showed a slightly higher incidence in GC heterozygotes than in GG homozygotes and clearly higher than in CC homozygotes (Figure 3B). Alternatively, we analyzed the results according to the frequency of GG, GC, and CC genotypes in the groups of patient samples classified by the presence of mutations at any of the three sites. Compared to the samples without mutation in the 3′UTR, the samples with mutation at any of the three sites showed a higher frequency of GC heterozygotes to the detriment of GG and CC homozygotes. Such enrichment in GC genotypes reached statistical significance for position +26 (Figure 3C). An analysis of allele frequencies for SNP rs8570 showed that such frequencies were not in Hardy–Weinberg equilibrium in both the group of samples with no mutations and the group of samples with mutations at position +26 (Figure 3D,E); the others were in equilibrium (not shown). The disequilibrium was due to enrichment for CC homozygotes in the unmutated group and enrichment for GC heterozygotes in the +26 mutation group.

The 3′UTR of mRNAs has a general function in regulating mRNA stability, nuclear export, and translation efficiency [21]. Since allelic composition at the rs8570 SNP position influences *RRAS2* mRNA abundance [17], we investigated whether the somatic mutations at any of the three positions found here influence mRNA expression. First, we found that the group of patient samples with no mutations had significantly higher mRNA expression than the group of patient samples with at least one mutation (Figure 3F). The site-specific analysis showed that a mutation at any of the three positions resulted in significantly lower mRNA abundance than the no mutation group (Figure 3G). These results show that the presence of mutations at any of the three positions in the 3′UTR inversely correlates with *RRAS2* mRNA expression.

### 3.3. Association of the Three 3′UTR Mutations with Clinical Traits

Overexpression of *RRAS2* mRNA is associated with a higher proportion of CLL cases at diagnosis versus pre-malignant monoclonal B-cell lymphocytosis (MBL), a higher proportion of leukemias bearing unmutated IgH gene, male sex, higher age, higher proportion of chromosomal anomalies, higher lymphocytosis, and lower platelet counts—all markers of poorer prognosis [22]. Therefore, we next determined if there was any association between the three somatic mutations in the 3′UTR of *RRAS2* and clinical data. We found that unlike for SNP rs8570, there was no significant difference in the percentage of patients with MBL versus full-blown CLL at diagnosis between unmutated and mutated samples at any of the three positions (Figure 4A). Likewise, we did not find significant differences in the distribution of mutated and unmutated samples according to the mutated/non-mutated IgH status (Figure 4B) or the existence of chromosomal aberrations by FISH (Figure 4C). There was slightly less male/female disequilibrium in patients with mutated versus unmutated 3′UTR, but it did not reach statistical significance (Figure 4D). Also, and unlike for SNP rs8570, there were not significant differences in age distribution (Figure 4E) or platelet counts (Figure 4F). Interestingly, we did find an inverse relationship between the presence of somatic mutations at any of the three positions in the 3′UTR and total leukocyte counts (Figure 4G), total lymphocyte counts (Figure 4H), and the percentage of CD19+CD5+ leukemic cells in blood (Figure 4I), compared to the samples with no mutations. These results suggest that malignant leukemic cell expansion in CLL patients is hampered by the acquisition of somatic mutations at any of the three sites of the 3′UTR of *RRAS2*.

## 4. Discussion

In this study, we describe three somatic mutations in the 3′UTR of *RRAS2* mRNA in CLL patients. In contrast to the rarity of missense mutations in the *RRAS2* gene, the 3′UTR harbors mutational hotspots at specific positions: +26 (G>T), +53 (T>C), and +180 (T>C) after the stop codon. Mutations within the 3′UTR of genes have historically been overshadowed by their counterparts within coding regions or canonical splice sites. However, the importance of these non-coding mutations is becoming increasingly evident, as they can disrupt RNA-protein interactions, microRNA binding sites, or other regulatory elements, thereby impacting gene expression and cellular phenotypes [23].

Our results reveal an intriguing relationship between *RRAS2* mRNA expression and the three 3′UTR somatic mutations. The newly identified 3′UTR mutations present an inverse relationship with *RRAS2* mRNA expression levels and correlate with more positive prognostic or less severe clinical factors (reduced lymphocytosis and improved platelet count). The reduction in *RRAS2* mRNA expression levels could be behind the reduced lymphocytosis observed in CLL patients with these mutations. Thus, the effect of the three somatic mutations at the 3′UTR seems to exert the opposite effect to the expression of the alternate C-allele at the SNP rs8570 [17]. Overall, our findings hint at a complex relationship between *RRAS2* mRNA expression, the rs8570 SNP, and the three somatic mutations identified in the gene’s 3′UTR. Higher *RRAS2* expression is linked to the dose of the C-allele: CC homozygous leukemias express more mRNA than GC heterozygous, and those more than GG homozygous. Comparing *RRAS2* mRNA expression between mutated and unmutated samples at any of the three 3′UTR positions for the three SNP genotypes, we show that mutation reduces *RRAS2* expression, but the reduction is especially significant for GC SNP heterozygotes. In terms of frequency, the presence of either one of the three somatic mutations or their combinations is also associated with GC heterozygosity at the SNP position. It can be hypothesized that the three somatic mutations in the 3′UTR are selected to limit the effect of the SNP C-allele on mRNA expression, and that such effect is in *cis*. This could be the reason that the mutations are preferentially selected in rs8570 heterozygote (GC) patients because they have just one chromosome to be modified. In fact, after sequencing the entire 3′UTR region expanding the first 200 nucleotides after the stop codon, we find a preferential enrichment in the three mutations in the chromosome bearing the C-allele compared to the one bearing the G-allele. This conclusion is preliminary, given the relatively small number of GC heterozygotes at the SNP position that we have been able to sequence. If confirmed, the association in *cis* could indicate that mutations in positions +26, +53, and +180 could interfere with microRNAs or other regulatory mechanisms associated with the SNP in position +124.

Our leading hypothesis consists of a compensatory mechanism at play. The 124 C-allele (rs8570) has been associated with higher *RRAS2* expression and, consequently, a worse disease prognosis [17]. We propose that the presence of three newly identified 3′UTR mutations compensates for the negative attributes of the 124 C-allele and could potentially reduce the adverse effects of highly elevated *RRAS2* expression. In addition to describing three somatic mutations in the 3′UTR sequence of the *RRAS2* gene, we reinforce the idea that the SNP rs8570, and overall *RRAS2* overexpression in its wild-type form, is linked to CLL development. The major finding, in addition to linking the alternate C-allele to higher mRNA expression, was that the distribution of GG, GC, and CC genotypes was tilted towards a higher abundance of CC homozygotes in the CLL patient cohort [17]. Here, we have increased the number of patient samples in the CLL cohort and compared it with a cohort of healthy blood donors. Thus, we confirm the disproportion in favor of CC homozygotes within the CLL cohort and show that such disproportion is not found in the healthy population cohort, which is in Hardy–Weinberg equilibrium. However, we still do not know if the increased frequency of CC homozygotes within the cohort of CLL leukemic cells is originated from somatic G>C mutation at the SNP position or if it is due to a higher propensity of the human population with CC homozygosity at the SNP position to develop CLL.

In the mouse model of *RRAS2* overexpression driving CLL development, we found that with time, there is a selection in favor of leukemic cells with even higher expression of *RRAS2* mRNA [17]. Therefore, a higher proportion of leukemias bearing the SNP rs8570 in CC homozygosity is understandable due to its association with higher expression. By contrast, what could be the reason for the emergence of the three somatic mutations described here that result in lower *RRAS2* expression and lower lymphocytosis? Our running hypothesis is that during the biological evolution of the leukemic CLL clones, excessive *RRAS2* overexpression could lead to senescence or to a dead-end differentiation of the pre-malignant B-cell clones. Overall, the results shown in this paper further link wild-type *RRAS2* overexpression to CLL development and clinical progression.

## 5. Conclusions

We have identified three mutational hotspots in the 3′ UTR of the *RRAS2* mRNA, affecting positions +25, +53, and +180 after the STOP codon, in 51% of our cohort of CLL patients. The presence of these three mutations alone or in combination results in reduced expression of *RRAS2* mRNA in CLL samples compared to those not bearing mutations. The presence of any of the three mutations inversely correlates with the number of leukocytes, total lymphocytes, and leukemic cells in the blood. The association of these three mutations with reduced *RRAS2* mRNA expression and diminished lymphocytosis reinforces the idea that *RRAS2* overexpression is implicated in CLL development and disease progression.

## Figures and Tables

**Figure 1 cells-12-02687-f001:**
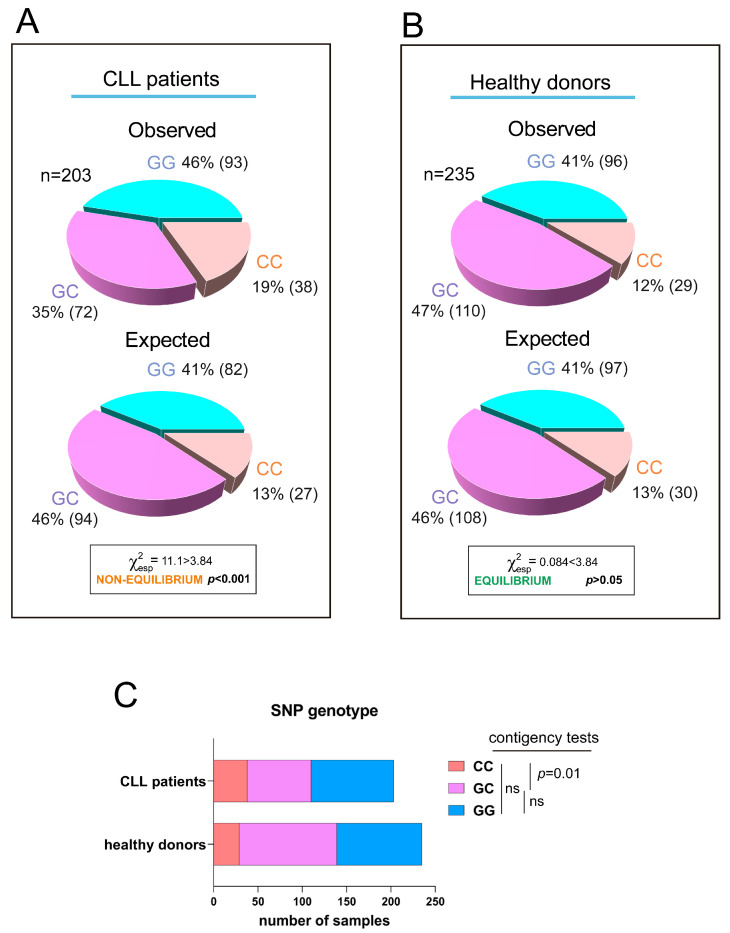
The C-allele at rs8570 SNP in the 3′UTR of *RRAS2* is overrepresented in CLL patients compared to the normal population. (**A**) Pie chart representation of the observed versus the expected distribution of the GG, GC, and CC genotypes at the rs8570 SNP in blood samples from our cohort of CLL patients (n = 203). The Chi-squared value (χ^2^) is greater than 3.84 such that the hypothesis that the observed and expected distributions are equivalent is rejected at *p* < 0.001. (**B**) Pie chart representation of the observed versus the expected distribution of the GG, GC, and CC genotypes at the rs8570 SNP in blood samples from our cohorts of healthy volunteers (n = 235). The Chi-squared value (χ^2^) is lower than 3.84 such that the hypothesis that the observed and expected distributions are different is rejected at *p* > 0.05. (**C**) Stacked bar plot showing a comparison of the number of blood samples with GG, GC, and CC genotypes at the rs8570 SNP within the patient and healthy donor cohorts. Contingency tests were carried to compare two-by-two the frequencies of genotypes between the patient and the healthy volunteer cohorts. A Chi-squared test was used to compute the *p* values. ns, not significant (*p* > 0.05).

**Figure 2 cells-12-02687-f002:**
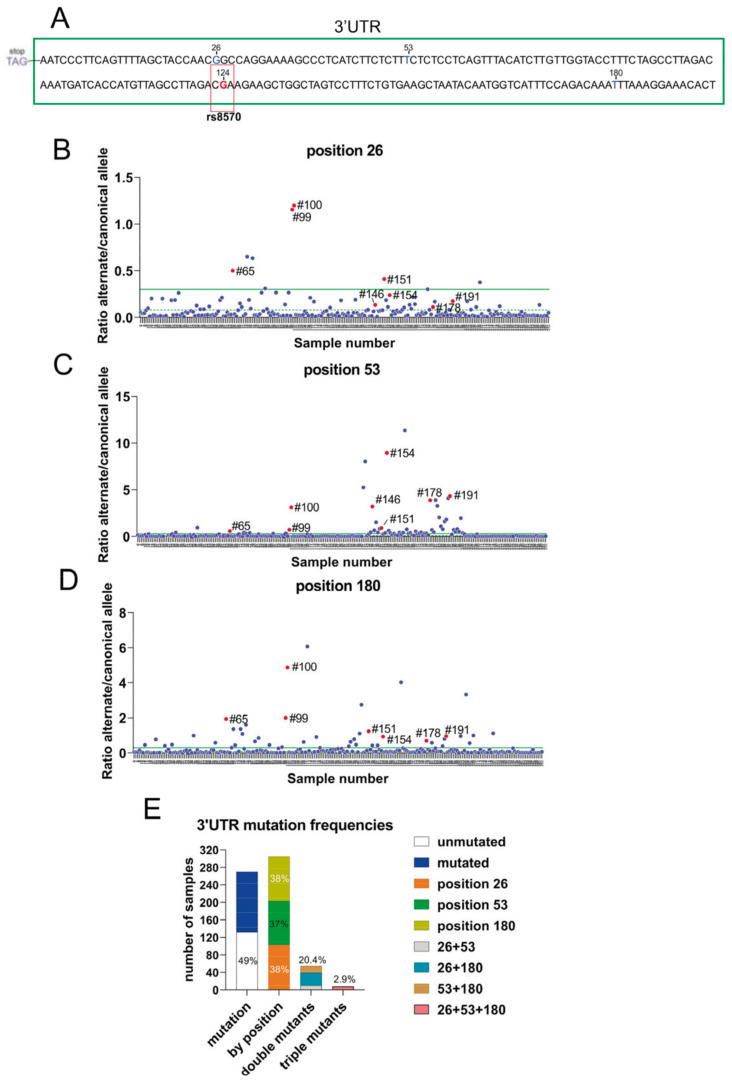
The 3′UTR of *RRAS2* frequently bears somatic mutations at positions +26, +53, and +180 from the stop codon. (**A**) Schematic representation of the site of rs8570 in the 3′UTR of *RRAS2* mRNA, 124 nucleotides downstream of the stop codon. The SNP position is indicated in a box and the canonical nucleotide (G) at that position is in red. Other sites in the 3′UTR (+26, +53, and +180) at which somatic mutations are found in our cohort of CLL patients are indicated with the canonical nucleotides. (**B**–**D**) Dedicated qPCRs were used to determine the ratio between the alternate and canonical nucleotides for each of the three sites found mutated in the cohort (n = 270) of CLL blood samples and represented in the y-axis. The sample number is indicated in the x-axis. The mean value of all ratios is indicated with a solid green line and the median with a broken green line. Samples with ratios above the median were considered as bearing the alternate allele at least in heterozygosity. Numbers beside the red dots point out to the samples containing simultaneously the three mutations. (**E**) Stacked bar plot showing the distribution of CLL blood samples according to the presence of somatic mutations at positions +26, +53, and +180 in isolation or in combinations.

**Figure 3 cells-12-02687-f003:**
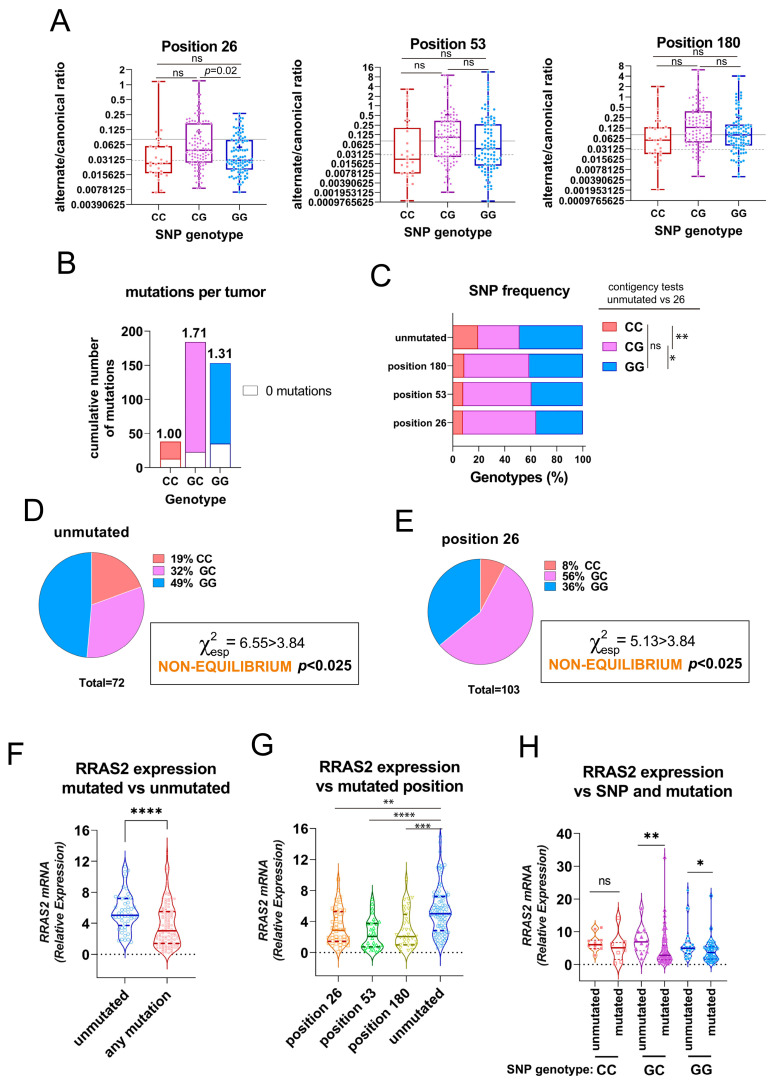
The three somatic mutations in the 3′UTR of *RRAS2* are associated with the SNP rs8570 in heterozygosity and lowering mRNA expression. (**A**) Box and whisker plots showing all the points and the median and the mean (cross, +) values of the alternate/canonical frequency ratios for each of the 3′UTR mutated positions, classified according to the SNP rs8570 genotype. Significance was assessed using a multiple comparison one-way ordinary ANOVA test. ns, not significant (*p* > 0.05). (**B**) Stacked bar plot showing the total number of mutations at the three 3′UTR positions in the patient CLL cohort classified according to the SNP rs8570 genotype. Inserted numbers indicate the average number of mutations detected per CLL sample. (**C**) Stacked bar plot showing the percentage of SNP rs8570 genotypes in CLL patients classified according to the presence of somatic mutations at the indicated 3′UTR positions. Contingency tests were carried out to compare two-by-two the frequencies of genotypes between the different mutant populations. A Chi-squared test with Yates’ correction was used to compute the *p* values. **, *p* = 0.005; *, *p* = 0.016; ns, not significant (*p* > 0.05). (**D**,**E**) Pie chart representation of the observed distribution of the GG, GC, and CC genotypes at the rs8570 SNP in blood samples of the cohorts of CLL patients bearing no mutations in the 3′UTR and bearing mutation at the +26 position. The Chi-squared value (χ^2^) is greater than 3.84 for both groups such that the hypothesis that genotypes are at Hardy–Weinberg equilibrium is rejected with *p* < 0.025. (**F**) Violin plot showing all experimental points, the median, and the upper and lower quartiles for *RRAS2* expression in all CLL patients in our cohort, classified according to the presence or not of any mutation in the 3′UTR. Significance was assessed using a non-parametric unpaired Mann–Whitney rank test (****, *p* < 0.0001). (**G**) Violin plot showing all experimental points, the median and the upper and lower quartiles for *RRAS2* expression in all CLL patients in our cohort classified according to the presence or not of mutations at the indicated positions of the 3′UTR. Significance was assessed using a non-parametric unpaired Dunn’s multiple comparisons test (****, *p* < 0.0001; ***, *p* = 0.0002; **, *p* = 0.0016). (**H**) Violin plot showing all experimental points, the median, and the upper and lower quartiles for *RRAS2* expression in all CLL patients in our cohort, classified according to the SNP rs8570 genotype and to the presence or not of mutations at the three positions of the 3′UTR. Significance was assessed using a non-parametric unpaired Mann–Whitney rank test (**, *p* = 0.0067; *, *p* = 0.017). ns, not significant (*p* > 0.05).

**Figure 4 cells-12-02687-f004:**
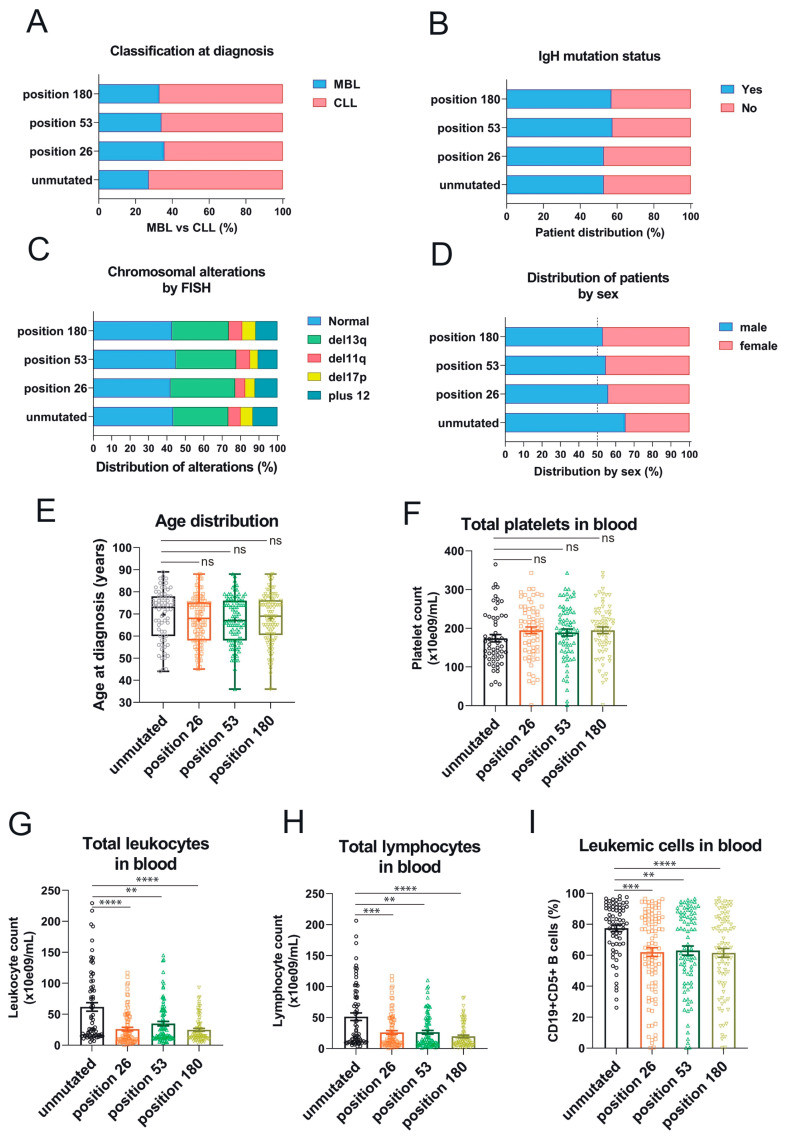
The three somatic mutations in the 3′UTR of *RRAS2* are associated with lower lymphocytosis in blood of CLL patients. (**A**) Stacked bar plot showing the percentage of patients with full-blown CLL or pre-malignant MBL at diagnosis, classified according to the presence or not of mutations at the indicated positions of the 3′UTR. Differences were not significant (*p* > 0.05) using an unpaired *t*-test. (**B**) Stacked bar plot showing the percentage of patients with leukemia bearing either mutated or unmutated *IgHV*, classified according to the presence or not of mutations at the indicated positions of the 3′UTR. Differences were not significant (*p* > 0.05) using an unpaired *t*-test. (**C**) Stacked bar plot showing the percentage of patients with leukemia bearing the indicated chromosomal anomalies and classified according to the presence or not of mutations at the indicated positions of the 3′UTR. Differences were not significant (*p* > 0.05) using a two-way ANOVA test. (**D**) Stacked bar plot showing the percentage of patients with leukemia classified according to their sex and to the presence or not of mutations at the indicated positions of the 3′UTR. Differences were not significant (*p* > 0.05) using contingency and Chi-squared tests. (**E**) Box and whisker plots showing all the points and the median and the mean (cross, +) values of the age at diagnosis for CLL patients bearing leukemias with mutations at the indicated positions. Significance was assessed using a non-parametric unpaired Dunn’s multiple comparisons test. ns, not significant (*p* > 0.05). (**F**) Scatter plot with bars showing all the points and the mean ± SEM values of the platelet concentration in blood of CLL patients. Significance was assessed using an ordinary one-way ANOVA test. ns, not significant (*p* > 0.05). (**G**,**H**) Scatter plots with bars showing all the points and the mean ± SEM values of the platelet concentration (**G**) and total lymphocytes (**H**) in blood of CLL patients. Significance was assessed using a non-parametric unpaired Dunn’s multiple comparisons test (****, *p* < 0.0001; ***, *p* = 0.0006; **, *p* = 0.0047). (**I**) Scatter plots with bars showing all the points and the mean ± SEM values of the percentage of leukemic CD19+CD5+ within blood lymphocytes in CLL patients. Significance was assessed using a non-parametric unpaired Dunn’s multiple comparisons test (****, *p* < 0.0001; ***, *p* = 0.0007; **, *p* = 0.0025).

**Table 1 cells-12-02687-t001:** PCR primers used for detection of the 3′UTR mutations.

3′UTR Mutation	Allele	Forward Primer Sequence	Reverse Primer Sequence
26	G (canonical)	CCCTTCAGTTTTAGCTACCCACG	GCTTCACAGAAAGGACTAGCC
	T (alternative)	CCCTTCAGTTTTAGCTACCCACT	
53	T (canonical)	GGAAAAGCCCTCATCTTCTATTT	AGCAGCCTTAGTGTTTCCTT
	C (alternative)	GGAAAAGCCCTCATCTTCTATTC	
124	G (canonical)	GATCACCATGTTAGCCTTATACC	AGCAGCCTTAGTGTTTCCTT
	C (alternative)	GATCACCATGTTAGCCTTATACG	
180	T (canonical)	CTACCAACGGCCAGGAAAAG	GCAGCCTTAGTGTTTCCTTGAAA
	C (alternative)		GCAGCCTTAGTGTTTCCTTGAAG

## Data Availability

The data presented in this study are available upon request to the corresponding author.

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
