# Peer review of "Characterization of Three Somatic Mutations in the 3′UTR of RRAS2 and Their Inverse Correlation with Lymphocytosis in Chronic Lymphocytic Leukemia"

_cells, 2023, doi:10.3390/cells12232687_

Round 1
Reviewer 1 Report
Comments and Suggestions for Authors
Lacuna et al, in the manuscript titled “Characterization of Three Somatic Mutations in the 3'UTR of RRAS2 and Their Clinical Implications in Chronic Lymphocytic Leukemia” describe three somatic mutations in the 3' untranslated region (3'UTR) affecting positions +26, +53, and +180 and their contribution to CLL pathology. This manuscript is a decent continuation of the group’s previous work https://www.ncbi.nlm.nih.gov/pmc/articles/PMC9913312/
Comments are noted below:
1. Are there clinical data regarding patients’ Ibrutinib response to RRAS2 mutations 245 at positions +26, +53, and +180 and SNP at rs8570?
2. Are there race and ethnicity correlations to the mutation abundance described in this manuscript?
3. 3’UTR tend to be conserved across organisms what are presumptive divers for these mutations?
4. Please provide possible explanations for the inverse relationship between CLL expansion and the mutations described in the text “Interestingly, we did find an inverse relationship between 319 the presence of somatic mutations at any of the three positions in the 3’-UTR and total 320 leukocyte counts (Fig. 4G), total lymphocyte counts (Fig. 4H), and the percentage of 321 CD19+CD5+ leukemic cells in blood (Fig. 4I), compared to the samples with no mutations. 322 These results suggest that malignant leukemic cell expansion in CLL patients is hampered 323 by the acquisition of somatic mutations at any of the three sites of the 3’-UTR of RRAS2”
Author Response
Lacuna et al, in the manuscript titled “Characterization of Three Somatic Mutations in the 3'UTR of RRAS2 and Their Clinical Implications in Chronic Lymphocytic Leukemia” describe three somatic mutations in the 3' untranslated region (3'UTR) affecting positions +26, +53, and +180 and their contribution to CLL pathology. This manuscript is a decent continuation of the group’s previous work https://www.ncbi.nlm.nih.gov/pmc/articles/PMC9913312/
Comments are noted below
- Are there clinical data regarding patients’ Ibrutinib response to RRAS2 mutations 245 at positions +26, +53, and +180 and SNP at rs8570? Patients with at least a C respond significantly better to ibrutinib in terms of reduced lymphocytosis, as shown in our previous article [1]. In regard to the three mutations in the 3’-UTR described here, we do not have unfortunately sufficient information as to provide a correlation with response to therapies.
- Are there race and ethnicity correlations to the mutation abundance described in this manuscript? All samples analyzed here are from Caucasian patients, so we cannot conclude if there is any differences in frequencies according to ethnicity. Furthermore, since these mutations are not described so far, this is a point worth analyzing in future research. We now, state in Methods that all patients are Caucasians.
- 3’UTR tend to be conserved across organisms what are presumptive divers for these mutations? We do not know. We speculate in the Discussion section that the mutations may be selected during the evolution of the clonal tumoral cells in order to avoid excessive expression of RRAS2 that could lead to terminal differentiation, senescence and a dead end. We postulate that there must be other compensatory mechanisms in the rest of the patients that do not bear mutations in the 3’-UTR to bypass such a dead end.
- Please provide possible explanations for the inverse relationship between CLL expansion and the mutations described in the text “Interestingly, we did find an inverse relationship between 319 the presence of somatic mutations at any of the three positions in the 3’-UTR and total 320 leukocyte counts (Fig. 4G), total lymphocyte counts (Fig. 4H), and the percentage of 321 CD19+CD5+ leukemic cells in blood (Fig. 4I), compared to the samples with no mutations. 322 These results suggest that malignant leukemic cell expansion in CLL patients is hampered 323 by the acquisition of somatic mutations at any of the three sites of the 3’-UTR of RRAS2” Yes, this is the puzzling observation. When we found the mutations, we were expecting that the mutations were going to promote higher overexpression of RRAS2, perhaps by limiting the effects of regulatory microRNAs, but this is not the case. The effect on expression is the opposite, and this correlates with lower lymphocyte counts in blood. Our explanation again is that there must be a selection for those mutations during the evolution and selection of the malignant clones. In any case, the inverse correlation between RRAS2 expression and lymphocytosis confirms the relationship between overexpression of this gene and malignancy in CLL.
Reviewer 2 Report
Comments and Suggestions for Authors
This manuscipt describes 3 new mutations in the UTR region of the RRAS2 gene in CLL patients.
the manuscript can be largely improved
the title is misleading since it mentions "their clinical implications" and should be more lighted up in the text.
Again the summary mentiones targeted therapies and there is no mention in the text (discussion)
Concerning the pronostic implication it should be mentioned: do we have a clinical implication of these 3 mutations?, all the data are available and if not another study is waranted
Fig1 is not needed since there is no mention of mutations
may be in supplementary data
what is the frequency in the normal population? (added to figure 2)
The figures should be reworked some data just mentioned in the text
the excel file given as supplementary data needs to be improved with
mean data extrems also (Max, minimun) and p value between healthy volonteers and patients.
minors comments:
page 2 Lane 92 the L after CLL should be removed
supplementary data: column Pathology should be replaced by clinical status
Author Response
This manuscipt describes 3 new mutations in the UTR region of the RRAS2 gene in CLL patients.the manuscript can be largely improved
the title is misleading since it mentions "their clinical implications" and should be more lighted up in the text.
We have now lighted up the title to make less aggressive in terms of clinical implications: “Characterization of Three Somatic Mutations in the 3'UTR of RRAS2 and their Inverse Correlation with Lymphocytosis in Chronic Lymphocytic Leukemia”
Again the summary mentiones targeted therapies and there is no mention in the text (discussion)
Sorry, we did not mean current targeted therapies but rather future targeted therapies aiming at inhibiting R-RAS2 itself.
Concerning the pronostic implication it should be mentioned: do we have a clinical implication of these 3 mutations?,
The clinical implications of these mutations are depicted in Figure 4 and are confined to their inverse correlation with the number of leukemic cells in the blood. A higher number of leukemic cells in the blood signifies a more advanced stage of the disease, as lymphocytosis is the primary parameter employed to evaluate disease progression. Apart from this, we lack sufficient data in our cohort to identify additional correlates of a worse prognosis.
all the data are available and if not another study is waranted
Sure, this is only the first paper describing the three mutations. Other must follow as soon as we gather additional information.
Fig1 is not needed since there is no mention of mutations may be in supplementary data
We disagree with the suggestion made by the reviewer. The foundation of this work lies in our prior discovery of an association between the SNP rs8570 in the 3'-UTR of RRAS2 and increased mRNA abundance, elevated lymphocytosis, and a direct correlation with other clinical parameters indicative of a worse prognosis [2]. In the original paper, we did not include a comparison of SNP allele frequencies in the patient and normal healthy population. We present these data here, and we believe that, in conjunction with the identification of three somatic mutations, they provide insights into the significance of this region of the RRAS2 mRNA in CLL.
what is the frequency in the normal population? (added to figure 2)
We have not sequenced the normal population, but since the 3 mutations are not described as Single Nucleotide Variants (SNV), their frequency in the normal population must be lower than 1%.
The figures should be reworked some data just mentioned in the text
the excel file given as supplementary data needs to be improved with
mean data extrems also (Max, minimun) and p value between healthy volonteers and patients.
We do not have data in terms of leukocyte numbers, etc for healthy volunteers and we do not have RRAS2 expression data for this cohort of healthy volunteers. RRAS2 mRNA expression was set as 1 from the mean expression in a cohort of healthy volunteers in our first publication [2]. However, we now do provide statistics of RRAS2 expression for the patient cohort. This is now included in Table S1.
minors comments:
page 2 Lane 92 the L after CLL should be removed
This has been corrected.
supplementary data: column Pathology should be replaced by clinical status
We do not agree with this. MBL and CLL are considered different pathologies and not a clinical status. MBL is considered a pre-malignant form of CLL. Some patients with MBL never evolve into CLL and others do. The classification of a lymphocytosis into MBL or CLL depends on the number of lymphocytes per mm3 in blood.
References
- Hortal, A.; Lacuna, M.; Cifuentes, C.; Alcoceba, M.; Bustelo, X.R.; González, M.; Alarcón, B. An Optimized Single Nucleotide Polymorphism-Based Detection Method Suggests That Allelic Variants in the 3’ Untranslated Region of RRAS2 Correlate with Treatment Response in Chronic Lymphocytic Leukemia Patients. Cancers 2023, 15, 644, doi:10.3390/cancers15030644.
- Hortal, A.M.; Oeste, C.L.; Cifuentes, C.; Alcoceba, M.; Fernández-Pisonero, I.; Clavaín, L.; Tercero, R.; Mendoza, P.; Domínguez, V.; García-Flores, M.; et al. Overexpression of Wild Type RRAS2, without Oncogenic Mutations, Drives Chronic Lymphocytic Leukemia. Mol. Cancer 2022, 21, 35, doi:10.1186/s12943-022-01496-x.
Round 2
Reviewer 2 Report
Comments and Suggestions for Authors
now the paper is ready fo publication